# Physical and Optical Properties of Aged Biomass Burning Aerosol from Wildfires in Siberia and the Western US at the Mt. Bachelor Observatory

James R. Laing[1], Daniel A. Jaffe[1,2], Jonathan R. Hee[1]

[1] University of Washington Bothell, Bothell, WA, USA
[2] University of Washington, Seattle, WA, USA

*Correspondence to*: Daniel A. Jaffe (djaffe@uw.edu)

**Abstract.** The summer of 2015 was an extreme forest fire year in the Pacific Northwest. Our sample site at the Mt. Bachelor Observatory (MBO, 2.7 km a.s.l.) in central Oregon observed biomass burning (BB) events more than 50% of the time during August. In this paper we characterize the aerosol physical and optical properties of 19 aged BB events during August 2015. Six of the nineteen events were influenced by Siberian fires originating near Lake Baikal

that were transported to MBO over 4-10 days. The remainder of the events resulted from wildfires in Northern California and Southwestern Oregon with transport times to MBO ranging from 3-35 hours. Fine particulate matter (PM1), carbon monoxide (CO), aerosol light scattering coefficients ($\sigma_{scat}$), aerosol light absorption coefficients ($\sigma_{abs}$), and aerosol number size distributions were measured throughout the campaign. We found that the Siberian events had a

significantly higher $\Delta\sigma_{abs}/\Delta CO$ enhancement ratio, higher mass absorption efficiency (MAE; $\Delta\sigma_{abs}/\Delta PM1$), lower single scattering albedo ($\omega$), and lower Absorption Ångström exponent (AAE) when compared with the regional events. We suggest that the observed Siberian events represent that portion of the plume that has hotter flaming fire conditions and thus enabled strong pyro-convective lofting and long-range transport to MBO. The Siberian events observed at MBO

therefore represent a selected portion of the original plume that would then have preferentially higher black carbon emissions and thus an enhancement in absorption. The lower AAE values in the Siberian events compared to regional events indicate a lack of brown carbon (BrC) production by the Siberian fires or a loss of BrC during transport. We found that mass scattering efficiencies (MSE) for the BB events ranged from 2.50-4.76 $m^2$ $g^{-1}$. We measured aerosol size

distributions with a scanning mobility particle sizer (SMPS). Number size distributions ranged from unimodal to bimodal and had geometric mean diameters ($D_{pm}$) ranging from 138-229 nm and geometric standard deviations ($\sigma_g$) ranging from 1.53-1.89. We found MSEs for BB events

to be positively correlated with the geometric mean of the aerosol size distributions ($R^2 = 0.73$), which agrees with Mie Theory. We did not find any dependence on event size distribution to transport time or fire source location.

## 1. Introduction

Biomass burning (BB) is a major source of aerosol in the atmosphere [*Andreae and Merlet*, 2001; *Bond et al.*, 2004]. BB particles are predominantly organic carbon (OC) and black carbon (BC), with some inorganic material [*Reid et al.*, 2005b; *Vakkari et al.*, 2014]. These particles can significantly impact the Earth's radiative balance and climate through direct and indirect aerosol effects. The direct effects on radiative forcing are due to the light scattering and absorption of the aerosol [*Boucher et al.*, 2013; *Haywood and Boucher*, 2000], and the indirect effects are caused by particles acting as cloud condensation nuclei (CCN) which affects cloud albedo [*Pierce et al.*, 2007; *Spracklen et al.*, 2011]. According to the IPCC 2013 report the largest uncertainty in determining global radiative forcing comes from quantifying the direct and indirect effects of aerosols [*Boucher et al.*, 2013]. Biomass burning is a major contributor to global aerosol burden and it has been predicted that these emissions are likely to increase due to climate change, particularly in the boreal forests of North America and Russia [*Flannigan et al.*, 2009; *Stocks et al.*, 1998] and in the western US [*Y Liu et al.*, 2014b; *Westerling et al.*, 2006]. This makes the proper characterization of aged BB emissions even more important.

Currently there are few field measurements of well-aged BB emissions. Our knowledge of BB aerosol primarily comes from laboratory experiments and near-field measurements taken within a few hours of a wildfire [*May et al.*, 2015; *May et al.*, 2014; *Okoshi et al.*, 2014; *Vakkari et al.*, 2014; *Yokelson et al.*, 2013b; *Yokelson et al.*, 2009]. *Holder et al.* [2016] showed that laboratory measurements of aerosol optical properties do not accurately reproduce field measurements. Freshly emitted BB particles are small in diameter (30-100 nm) [*Hosseini et al.*, 2010; *Levin et al.*, 2010]. As the plume ages, the aerosol undergoes rapid chemical and physical changes on the time scale of minutes to hours [*Reid et al.*, 2005a; *Reid et al.*, 2005b; *Vakkari et al.*, 2014]. The change in particle size is due to coagulation and the condensation of organic material onto the existing particles [*Reid et al.*, 2005b; *Seinfeld and Pandis*, 2006]. The coagulation rate can be very high in fresh BB plumes since this is equivalent to the square of

particle number concentration. This process increases the size of the particles while decreasing the number concentration. Condensation of secondary organic aerosol (SOA) onto particles in BB plumes also increases particle size. The condensation of SOA is counterbalanced by the loss

of primary organic aerosol (POA), which can evaporate during plume dilution [*May et al.*, 2015; *May et al.*, 2013]. The net condensation/evaporation effect is highly variable. Some studies have observed an increase in mass with plume age due to SOA production [*Briggs et al.*, 2016; *Hobbs*, 2003; *Vakkari et al.*, 2014; *Yokelson et al.*, 2009], while others have observed limited SOA formation [*Akagi et al.*, 2012; *Jolleys et al.*, 2015]. All of these uncertainties in the aging process

of biomass burning underscores the importance of characterizing the physical and optical properties of well-aged biomass burning aerosol.

In this study we analyze 19 aged BB events observed in the summer of 2015 at Mt. Bachelor in Oregon. The BB events consisted of Regional events (fires in Northern California and Southwestern Oregon; transported 3-35 hours) and Siberian fire events (fires around Lake

Baikal; transported 4-10 days). We investigated the aerosol optical and physical properties of these events and explored their variation with source location and transport time. This study addresses the following questions:

- What are the differences in the optical properties of regional and Siberian BB events observed at MBO?

- What is the range of mass scattering efficiencies for BB events and what explains their variability?

- What is the range in aerosol size distributions of BB events at MBO and how does this vary with plume age?

## 2. Methods

### 2.1. Mt. Bachelor Observatory

The Mt. Bachelor Observatory (MBO) is a mountaintop site that has been in operation since 2004 [*Jaffe et al.*, 2005] . It is located at the summit of Mt. Bachelor in central Oregon, US (43.98° N, 121.69° W, 2,764 m a.s.l.). A suite of measurements (including carbon monoxide (CO), ozone ($O_3$), aerosol scattering coefficients, and more) have been made continuously at the

summit site. Previous studies have observed BB plumes in the free troposphere from regional

and distant sources in the spring, summer, and fall [*Baylon et al.*, 2015; *Briggs et al.*, 2016; *Collier et al.*, 2016; *Timonen et al.*, 2014; *Weiss-Penzias et al.*, 2007; *Wigder et al.*, 2013], and long-range transport of Asian pollution in the spring [*Ambrose et al.*, 2011; *Fischer et al.*, 2010a; *Fischer et al.*, 2010b; *Gratz et al.*, 2014; *Jaffe et al.*, 2005; *Reidmiller et al.*, 2010; *Timonen et al.*, 2014; *Timonen et al.*, 2013; *Weiss-Penzias et al.*, 2006]. During the summer of 2015 an intensive field campaign was performed at MBO to measure aerosol physical and optical properties of wildfire emissions.

**2.2. CO, $CO_2$, and Meteorological Data**

CO and $CO_2$ measurements were made using a Picarro G2302 Cavity Ring-Down Spectrometer. Calibrations were performed every 8 hours using three different NOAA calibration gas standards, which are referenced to the World Meteorological Organization's (WMO) mole fraction calibration scale [*Gratz et al.*, 2014]. Total uncertainty based on the precision of calibrations over the campaign was 3%. Basic meteorology measurements, such as temperature, humidity and wind speed were also measured continuously [*Ambrose et al.*, 2011].

**2.3. Aerosol Instruments**

We measured dry (relative humidity (RH) less than 35%) aerosol scattering and absorption coefficients, aerosol number size distribution, and particle mass during the 2015 summer campaign in 5 minute averages. An inline 1 μm impactor was located prior to the aerosol instruments. The aerosol instruments were located in a temperature-controlled room within the summit building, situated approximately 15 m below the inlet. The aerosol sample line was situated such that the last 2.5 m was located within a space that was temperature controlled at 20 ± 3°C, typically 10°C–20°C warmer than ambient. Relative humidity of the sampled air was less than 35% throughout the campaign. The temperature increase from going outside into the heated building reduced the RH of the sample. RH was measured in the sample airstream by the nephelometer and SMPS. The average RH during the campaign measured by the nephelometer and SMPS was 22.1% and 22.6%, respectively. Ninety-five percent of the 5 minute averaged samples had an RH less than 30%.

We measured multi-wavelength aerosol light scattering coefficients ($\sigma_{scat}$) using an integrating nephelometer (model 3563, TSI Inc., Shoreview, MN) at wavelengths 450, 550, and

700 nm. During the 2015 campaign the TSI nephelometer was periodically switched to measure

both particle free air and $CO_2$. The measured values were corrected for offset and calibration

drift in addition to angular nonidealities [*Anderson and Ogren*, 1998]. The filtered air and $CO_2$

were measured approximately every two weeks [*Anderson and Ogren*, 1998]. The data reduction

and uncertainty analysis that we followed for the scattering data are outlined by *Anderson and*

*Ogren* [1998]. Sources of uncertainties associated with the nephelometer include photon

counting noise, zeroing and calibration, and the correction for angular nonidealities. Combined

these uncertainties yielded total uncertainties of ~15% during BB events.

We measured aerosol light absorption coefficients ($\sigma_{abs}$) with a 3λ tricolor absorption

photometer (TAP, Brechtel Inc., Hayward, CA) at wavelengths 467, 528, and 660 nm.

Throughout the paper $\sigma_{scat}$ and $\sigma_{abs}$ values represent measurements taken at 550 nm and 528 nm,

respectively. The TAP is a new instrument that uses the same operating principle as the Particle

Soot Absorption Photometer (PSAP) and the same filters (47 mm PALL E70-2075W). Unlike

the PSAP, the TAP rotates through 8 filter spots per individual filter along with two reference

spots. During deployment at MBO, the TAP was set to rotate to the next filter spot when a filter

spot's transmission reached 50%. The absorption coefficients were corrected using the filter

loading and aerosol scattering correction factors derived for the 3λ PSAP by *Virkkula* [2010].

Uncertainty calculations were based on those used in a previous study at MBO for measurements

with a 3λ PSAP [*Fischer et al.*, 2010a]. Sources of uncertainty include noise, instrument drift,

errors in the loading function, the correction for the scattering artifact, and uncertainty in the

flow and spot size corrections [*Anderson et al.*, 1999; *Bond et al.*, 1999; *Virkkula et al.*, 2005].

Combining these uncertainties yielded total uncertainties of ~25-40% during BB events.

Single scattering albedo (ω) for each event was calculated as the Reduced Major Axis

(RMA) regression of scattering and total extinction (scattering + absorption) coefficient at 528

nm. To adjust the $\sigma_{scat}$ value from 550 nm to 528 nm, a power law relationship was assumed

between scattering and wavelength. The 450-550 nm pair was used to adjust the 550 nm $\sigma_{scat}$

measurement to 528 nm using equation (1):

$$\sigma\,{}^{528}_{scat} = \sigma\,{}^{550}_{scat} * \left(\frac{\lambda_{550}}{\lambda_{528}}\right)^{SAE_{450,550}} \tag{1}$$

where λ is wavelength and SAE is the scattering Ångström exponent calculated with the two

wavelengths specified. The SAE values were calculated for each 5-minute interval using the

scattering coefficients measured at 450 nm and 550 nm. Mean SAE values for the BB plumes

ranged from 1.61 to 2.15. Uncertainties for ω were calculated the same as the enhancement

ratios, which is discussed in Section 2.4.

Absorption Ångström exponent (AAE) values were calculated for the absorption

coefficient pair of 467 and 660 nm using equation (2):

$AAE = -log(\sigma_{abs}^{467}/\sigma_{abs}^{660})/log(467/660)$                       (2)

Uncertainties for AAE values were calculated by propagating the uncertainties from the

measurements used to calculate AAE using addition in quadrature [*Fischer et al.*, 2010a].

We measured 5-minute averaged dry aerosol number size distribution with a TSI 3938

Scanning Mobility Particle Sizer (SMPS). The SMPS system consisted of a TSI 3082

electrostatic classifier with a TSI 3081 Differential Mobility Analyzer (DMA) and a TSI 3787

water-based condensation particle counter. A total of 107 bins were used to measure a diameter

range from 14.1-637.8 nm. A sheath to aerosol flow ratio of 10:1 was used for the DMA. A

multiple charge correction and diffusion loss correction were applied to the SMPS particle

number concentration data using the TSI software. An additional diffusion correction for the

inlet tube (15 m, 12 LPM) was applied assuming a laminar flow [*Hinds*, 1999]. Prior to

deployment we confirmed the sizing accuracy of the SMPS using polystyrene latex spheres

(PSL).

We measured dry particle mass under 1 µm (PM1) with an Optical Particle Counter

(OPC, model 1.109, Grimm Technologies, Douglasville, GA). This is a U.S. EPA equivalent

method for measuring PM2.5 mass concentration. The OPC was factory calibrated prior to

deployment.

All particle measurements ($\sigma_{scat}$, $\sigma_{abs}$, PM1, number size distribution) were corrected to

standard temperature and pressure (STP; T = 273.15, P = 101.325 kPa).

### 2.4. Enhancement Ratio Calculations

Enhancement ratios ($\Delta Y/\Delta X$) were calculated from the slope of the RMA regression of Y plotted

against X. *Briggs et al.* [2016] calculated enhancement ratios (ERs) of BB plumes using three

different methods, one method using the RMA slope of the linear correlation of two species, and

two others calculating absolute enhancement above local background using two different

definitions of background. All three methods produced similar results for $\Delta\sigma_{scat}/\Delta CO$,

$\Delta NO_y/\Delta CO$, and $PAN/\Delta CO$, but differing results for $\Delta O_3/\Delta CO$. In our study we used the RMA

regression method for calculating ERs of $\Delta\sigma_{scat}/\Delta CO$ and $\Delta\sigma_{abs}/\Delta CO$.

       Mass scattering and mass absorption efficiencies (MSE and MAE) were calculated as the

enhancement ratios of $\Delta\sigma_{scat}/\Delta PM1$ and $\Delta\sigma_{abs}/\Delta PM1$, respectively, at 550 nm for $\sigma_{scat}$ and 528

nm for $\sigma_{abs}$. As previously mentioned, $\omega$ was calculated as the RMA regression of scattering and

total extinction (scattering + absorption). In all cases the enhancements ($\Delta$) are large compared to

background, thus avoiding the problems described by *Briggs et al.* [2016] for small

enhancements above background.

       We determined the uncertainties for the enhancement ratio calculations from the

uncertainties in the extensive properties used in calculating the enhancement ratios and the

uncertainty of the RMA regression using addition in quadrature. For example, the uncertainty in

$\Delta X/\Delta Y$ was calculated by adding in quadrature the uncertainty in the RMA regression, the

uncertainty in the X measurement, and the uncertainty in the Y measurement.

       We present both precision uncertainty and total uncertainty as described by [*Anderson et*

*al.*, 1999] for all values derived from optical measurements. Precision uncertainty includes

uncertainty associated with noise and instrument drift. This is best used when comparing

measurements collected using the same instruments and protocols. It is the appropriate

uncertainty to consider when comparing individual BB events seen at MBO in this study. Total

uncertainty includes precision uncertainty, the uncertainty associated with the corrections we

applied to the data, and the uncertainty associated with the calibration method. This is the

appropriate uncertainty to consider when comparing the measurements presented in this study

with data collected using other measurement methods.

### 2.5. Biomass Burning Event Identification

We identified BB events as time periods during which 5-min ambient aerosol scattering

coefficients $\sigma_{scat} > 20$ Mm$^{-1}$ for at least one hour, 5-min CO > 150 ppbv for at least one hour, and

there was a strong correlation ($R^2 > 0.80$) between $\sigma_{scat}$ and CO. To determine fire locations we

calculated back-trajectories using the National Oceanic and Atmospheric Administration Hybrid

Single-Particle Lagrangian Integrated Trajectory (HYSPLIT) model, version 4 [*Draxler*, 1999;

*Draxler and Hess*, 1997; 1998; *Stein et al.*, 2015]. We used the Global Data Assimilation System

(GDAS) $1° \times 1°$ gridded meteorological data from the National Oceanographic and Atmospheric

Administration's Air Resources Laboratory (NOAA-ARL). Within GDAS, the grid containing

MBO is located at ~1500 m amgl (above model ground level) so back-trajectory starting heights

of 1300, 1500, and 1700 m amgl were chosen [*Ambrose et al.*, 2011]. We identified fire locations

using Moderate Resolution Imaging Spectroradiometer (MODIS) satellite-derived active fire

counts [*Justice et al.*, 2002], and Fire INventory from NCAR (FINN) data [*Wiedinmyer et al.,*

*2011*]. Similar criteria for identifying BB events has been used by *Baylon et al.* [2015] and

*Wigder et al.* [2013] from data collected at MBO.

## 3.  Results and Discussion

### 3.1. Identified BB Events and Fire Source Identification

The summer of 2015 was a very active fire season in the Pacific Northwest (Figure 1). During

the month of August 2015, 51% of the 5-minute averages met the criteria for a BB event, having

$\sigma_{scat} > 20$ Mm$^{-1}$ and CO > 150 ppbv, including several multi-day periods. We split these multi-

day events up if discernable plumes within the event could be identified. Altogether we

identified 19 events, ranging from 1.5-45 hours in duration. We use the term *event*, not *plume*,

because of the long duration of some of the events and the fact that most BB events observed in

2015 were influenced by emissions from multiple fires.

Two large multi-day events of regional BB smoke from fires in Northern California and

Southwestern Oregon dominated the sampling period (dotted boxes in Figure 2). Transport time

from these regional fires to MBO, estimated from the back-trajectories, ranged from 3-35 hours.

In between these two large regional BB events there was a time period that was influenced by

Siberian wildfires (solid box in Figure 2). During August there were intense forest fires around

Lake Baikal in Siberia, peaking on 8/8/2015 with a total fire area of 681 km$^2$, and an estimated

CO and BC emissions of $3.22 \times 10^8$ and $1.33 \times 10^6$ kg/day, respectively (*Fire INventory from*

*NCAR (FINN) data) [Wiedinmyer et al., 2011]*. Transport times from northeastern Asia to MBO

during these events ranged from 4-10 days. NASA MODIS aqua and terra images show the

eastward transport of smoke from the Lake Baikal fires during this time period

[*https://worldview.earthdata.nasa.gov/*, 2016]. We used V3.30 aerosol classification products

from the Cloud-Aerosol Lidar with Orthogonal Polarization (CALIOP) instrument on the Cloud-

Aerosol Lidar Infrared Pathfinder Satellite Observation (CALIPSO) satellite to confirm the

transport of plumes of smoke from the Siberian fires to North America [*http://www-calipso.larc.nasa.gov/*, 2016; *Winker et al.*, 2010; *Winker et al.*, 2009]. Aerosol plumes are identified as one of six types: dust, polluted continental, polluted dust, smoke (biomass burning), clean continental or clean marine aerosols [*Omar et al.*, 2009].

### 3.2. Overview of Summer 2015 BB Events

Table 1 provides an overview of the 19 BB events from MBO during the summer of 2015. We calculated water vapor enhancement ($\Delta WV$) to indicate the origin of the event air mass. Positive $\Delta WV$ suggest the air mass ascended from the boundary layer (BL) to MBO, while near zero or negative values mean the air mass is relatively dry and likely descended or arrived from the free troposphere (FT) [*Baylon et al.*, 2015; *Wigder et al.*, 2013]. All of the regional BB events have $\Delta WV$ values $\geq 1.00$ g/kg, while all of the Siberian-influenced events have $\Delta WV$ values near zero or negative. The precision and total uncertainties for all of the parameters derived from optical measurements are provided for these events in Table S1.

We found the $\Delta\sigma_{scat}/\Delta CO$ ($\sigma_{scat}$ at STP) enhancement ratio to range from 0.48-1.29 Mm$^{-1}$ ppbv$^{-1}$, with the majority of events being between 0.8 and 1.25 Mm$^{-1}$ ppbv$^{-1}$. We found $\Delta PM1/\Delta CO$ (PM1 at STP) to range from 0.18-0.43 µg cm$^{-3}$ ppbv$^{-1}$. These values are in the same range as BB plumes seen previously at MBO [*Baylon et al.*, 2015; *Wigder et al.*, 2013].

In 2015 many fires were burning throughout the northwestern U.S. So in contrast to previous work at MBO, we were not able to calculate transport time for any of the regional BB events observed as they were influenced by multiple fires with various transport times. Figure 3 provides an example of this and exemplifies the impossibility of determining an exact transport time.

### 3.3. Optical Properties of the BB Aerosol at MBO

We observed significant differences in the optical properties of regional and Siberian-influenced BB events. The Siberian-influenced events had higher absorption coefficients relative to other measurements. This resulted in higher $\Delta\sigma_{abs}/\Delta CO$, higher MAE ($\Delta\sigma_{abs}/\Delta PM1$), and lower $\omega$ ($\sigma_{scat}/(\sigma_{scat} + \sigma_{abs})$) compared to regional BB events (Figures 4 and 5). We found no significant differences for $\Delta\sigma_{scat}/\Delta CO$ or MSE ($\Delta\sigma_{scat}/\Delta PM1$) between regional and Siberian events. Back-trajectories for the Siberian events (events 10-15) originated at high elevation over Siberia,

suggesting that the BB emissions were lofted to altitudes of 4-10 km (Figure 6). The Siberian events at MBO were observed over the course of a week (8/17/2015-8/23/2015); therefore the back-trajectories in Figure 6 represent a sustained meteorological pattern that consistently transported Siberian smoke to North America throughout the week. Aerosol vertical profiles measured by CALIOP corroborate the transport of BB plumes from the Siberian fires across the Pacific at altitudes of 4-10 km. Large BB plumes were identified over Northeast Asian and the North Pacific consisting primarily of BB smoke and some polluted dust over the Northern Pacific from 8/8/2015-8/17/2015. Figures S1-S4 show selected CALIPSO transects from 8/13/2015-8/16/2015 over the Pacific. The location and altitude of these plumes match the back-trajectories calculated from MBO for the Siberian events (Figure 6), verifying that events 10-15 are heavily influenced by the Siberian fires.

We suggest that the Siberian BB events observed at MBO represent hotter, more flaming portions of the fires which have higher BC emissions and thus higher absorption enhancements compared to the regional BB events. The hotter parts of the fires have more pyro-convective energy to loft the plume high into the atmosphere where it can then undergo long-range transport. During the ARCTAS-A flight campaign in Alaska, Siberian fire plumes were found to have a much larger BC/CO ratio ($8.5 \pm 5.4$ ng m$^{-3}$ ppbv$^{-1}$) than North American fire plumes ($2.3 \pm 2.2$ ng m$^{-3}$ ppbv$^{-1}$) [*Kondo et al.*, 2011]. This difference was attributed to the Siberian fires having a higher modified combustion efficiency (MCE). In addition, for the Siberian BB plumes they found MCE to increase with altitude. *Jolleys et al.* [2015] correspondingly found higher $\Delta$BC/$\Delta$OA ($\Delta$black carbon/$\Delta$organic aerosol) ratios to increase with altitude in eastern Canadian BB plumes. Intense, flaming fire plumes have higher injection heights into the atmosphere due to enhanced pyro-convection, whereas smoldering fires have low thermal convective energy and are mostly contained within the boundary layer. BB aerosol lofted to the free troposphere via pyro-convection is less likely to be removed and can have a longer atmospheric lifetime of up to 40 days [*Bond et al.*, 2013]. The back-trajectories for the Siberian events corroborate this idea. They were all relatively dry (water vapor mixing ratio < 5 g kg$^{-1}$) with little precipitation during transport, suggesting the aerosol in the Siberian events was subjected to very limited wet deposition, which is the main removal mechanism from the atmosphere. Flaming conditions produce more BC and less OA generally, which leads to amplified absorption [*Vakkari et al.*, 2014; *Yokelson et al.*, 2009]. Flaming conditions are associated with high MCE values [*Reid et*

*al.*, 2005a]. Unfortunately, we were not able to calculate MCE values for the Siberian events at

MBO due to extensive dilution and boundary layer mixing during transport [*Yokelson et al.*, 2013a].

While the ω values for the Siberian events are significantly lower relative to the regional events, they are all high (> 0.95) compared to typical flaming conditions measured in the laboratory or near-field measurements [*S Liu et al.*, 2014a; *Vakkari et al.*, 2014]. *S Liu et al.*

[2014a] found a robust relationship between ω and MCE in laboratory BB emissions where MCE was negatively correlated with ω. However, observations have found that ω increases significantly after emission in BB plumes [*Reid et al.*, 2005a; *Vakkari et al.*, 2014]. A previous study at MBO found that well-aged BB plumes do not follow the *S Liu et al.* [2014a] parametrization [*Briggs et al.*, 2016]. All of the BB plumes observed by *Briggs et al.* [2016] had

ω>0.91 despite MCE values as high as 0.98, and no relationship was found between ω and MCE. The high ω values typical of aged BB plumes are most likely due to SOA formation and increased scattering efficiency as the particles ages and increases in size through coagulation and condensation. Given this we believe the ω's seen in these Siberian plumes are different and significantly higher than the ω's directly after emission and are therefore cannot be equated to an

MCE value.

We found AAE values for the Siberian events to be significantly lower than regional BB events (Figures 4 and 5). High AAE values are indicative of the presence of brown carbon (BrC). Brown carbon is a fraction of OA that selectively absorbs short wavelengths [*Andreae and Gelencser*, 2006; *Chen and Bond*, 2010; *Kirchstetter et al.*, 2004]. There are two possible

explanations for the difference in AAE values. The first is that the flaming conditions that produced the Siberian events seen at MBO had higher BC and lower OA emissions, which inherently have lower AAE as total absorption is dominated by BC and less BrC is initially produced. Laboratory and field studies have corroborated this and observed an inverse relationship between MCE and AAE [*Holder et al.*, 2016; *S Liu et al.*, 2014a; *McMeeking et al.*,

2014]. The other explanation is that BrC is lost during transport through photobleaching, volatilization, and aerosol-phase reactions. *Forrister et al.* [2015] determined that BrC decreased with transport with a half-life of 9 hours and that AAE decreases from ~4.0 to ~2.5 24 hours after emission. All of the regional BB events were influenced by multiple fires that had transport times

varying from 3- 35 hours. With each event being influenced by at least one fire with a transport

time ≤ 6 hours, this short transport time is consistent with the higher AAE values we observed.

### 3.4. Mass Scattering Efficiency

Mass scattering efficiencies (MSEs) are important for calculating the radiative forcing effects of aerosols in global climate and chemical transport models. Estimates of MSE are used to convert aerosol mass measurements to aerosol optical properties [*Briggs et al.*, 2016; *Hand and Malm*,

2007; *Pitchford et al.*, 2007]. MSE is dependent on particle composition, which determines the particle's refractive index and hygroscopicity, and aerosol size distribution [*Hand and Malm*, 2007]. We calculated MSE as the slope of the RMA regression of $\sigma_{scat}$ and PM1 ($\Delta\sigma_{scat}/\Delta PM1$). $R^2$ values were >0.94 for all events. We found MSE values ranged from 2.50-4.76 $m^2$ $g^{-1}$, which are consistent with previously measured values.

During 2013 at MBO, MSE values estimated using AMS organic matter (OM) data and the $\sigma_{scat}$ for four wildfire plumes ranged from 2.8-4.8 $m^2$ $g^{-1}$ (mean: 3.7 $m^2$ $g^{-1}$) [*Briggs et al.*, 2016]. *Levin et al.* [2010] calculated MSE values for fresh BB smoke from a variety of fuels to range from 1.5–5.7 $m^2$ $g^{-1}$, with most of the values falling between 2.0 and 4.5 $m^2$ $g^{-1}$. *Reid et al.* [2005a] reviewed MSE values from BB events and found a range between 3.2 and 4.2 $m^2$ $g^{-1}$

(mean: 3.8 $m^2$ $g^{-1}$) for temperate and boreal fresh smoke, and larger values for aged smoke (3.5-4.6 $m^2$ $g^{-1}$; mean: 4.3 $m^2$ $g^{-1}$). MSE values upwards of ~6 $m^2$ $g^{-1}$ have been observed for aged BB plumes [*Hand and Malm*, 2007; *McMeeking et al.*, 2005]. Due to the large variation in MSE values for BB events, assigning an average MSE value to convert aerosol mass measurements to aerosol optical properties or vice versa introduces significant uncertainties.

We investigated the cause for the variation in the MSE values that we observed. We found MSE's for BB events to be positively correlated with $D_{pm}$ ($R^2 = 0.73$) (Figure 7a). If two $D_{pm}$ values associated with bimodal size distributions are removed, the correlation increases substantially ($R^2 = 0.88$). A positive correlation between MSE and mean particle diameter has previously been observed in ambient data [*Lowenthal and Kumar*, 2004] and laboratory studies

[*McMeeking et al.*, 2005]. Theoretically according to Mie theory, MSE will increases as the average particle diameter grows, through coagulation and condensation, toward the measurement wavelength (550 nm) [*Seinfeld and Pandis*, 2006].

### 3.5. BB Size Distributions

Figure 8 shows the BB aerosol number size distributions for the events we observed at MBO in
dN/dlogDp. We found $D_{pm}$ and $\sigma_g$ of the number distributions to range from 138-229 nm and
1.53-1.89, respectively. The size distributions observed at MBO are similar to *Janhäll et al.*
[2010], who compiled aged BB size distributions. They found the accumulation mode mean
diameter to range from 175-300 nm with geometric standard deviations of 1.3–1.7. No
dependence was found in $D_{pm}$ in plumes of regional or Siberian origins. Similarly during the
ARCTAS-B flight campaign, aged BB plumes of western Canadian and Asian origins were
found to have similar size distributions (Canadian: $D_{pm} = 224 \pm 14$ nm, $\sigma_g = 1.31 \pm 0.05$; Asian:
$D_{pm} = 238 \pm 11$ nm, $\sigma_g = 1.31 \pm 0.03$) [*Kondo et al.*, 2011]. The BORTAS-B flight campaign in
Eastern Canada observed aged BB plumes with median diameters of 180-240 nm [*Sakamoto et
al.*, 2015].
We observed clear bimodal distributions with an accumulation mode (100-500 nm) and
Aitken mode (20-100 nm) for five events (2, 3, 11, 14 and 15). The Aitken mode in these size
distributions most likely represents a secondary source from within the boundary layer. A
prominent "tail" consisting of higher than expected number concentrations of small-diameter
particles (30-90 nm) was observed for most of the unimodal events at MBO. It would be
expected that particles in this size range would grow to larger particles through coagulation
relatively quickly. *Sakamoto et al.* [2015] observed a similar elevation in the number
concentration of small particles during the BORTAS-B campaign. They attempted to account for
the existence of the tail with a Lagrangian box model of coagulation and dilution but were unable
to do so. Coagulation should cause a significant decrease in Aitken mode particles in a matter of
hours; and nucleation and condensation growth rates would have to be unreasonably high to
maintain these small particles.
        We observed no clear distinction between the size distributions from regional and
Siberian events. These results are consistent with previous studies that have not observed a
dependence from plume age, transport time, or source location on the BB size distribution.
*Kondo et al.* [2011] found little difference between the $D_{pm}$ of Siberian and Canadian BB plumes
despite different chemical composition, optical properties, and transport times. Similarly,
*Sakamoto et al.* [2015] found no trend in size distribution with plume transport distance. In a

study performed in the Front Range of Colorado, *Carrico et al.* [2016] found no significant difference between the size distribution of an hours old and a days old fire plume.

As previously stated, we found MSE's for BB events to be positively correlated with $D_{pm}$. This makes physical sense due to increased light scattering efficiency of larger particles closer to the wavelength of light (550 nm). In addition, we found event integrated $D_{pm}$ to be correlated with event integrated $\sigma_{scat}$ ($R^2 = 0.65$) and PM1 mass ($R^2 = 0.72$), and moderately correlated with CO ($R^2 = 0.41$) (Figure 7b,c,d). $D_{pm}$ was not found to be correlated with any

normalized enhancement ratio ($\Delta\sigma_{scat}/\Delta CO$, $\Delta PM1/\Delta CO$). CO, $\sigma_{scat}$, and PM1 can be thought of as surrogates for plume concentration. The correlation between these proxies of plume concentration and $D_{pm}$ indicates that in general, the more concentrated BB plumes have larger size distributions.

In a related study, *Sakamoto et al.* [2016] selected subsets of the MBO BB regional

events presented here and tested them against parameterizations of the aged size distribution. The parameterizations calculate $D_{pm}$ and $\sigma_g$ from inputs that can be derived from emissions-inventory and meteorological parameters. The seven inputs are: emission median dry diameter, emission distribution modal width, mass emissions flux, fire area, mean boundary-layer wind speed, plume mixing depth, and time/distance since emission. We identified eleven plumes from

regional events that had consistent transport to known regional fires. The simple fits captured over half of the variability in observed $D_{pm}$ and modal width, even though the freshly emitted $D_{pm}$ and modal widths were unknown. The results demonstrate that the parameterizations presented in *Sakamoto et al.* [2016] section 3.4 can be successfully used to estimate aged BB size distributions in regional BB plumes with transport times up to 35 hours. Using these

parameterizations to estimate BB plume size distribution in global and regional aerosol models is a significant improvement to assuming fixed values for size-distribution parameters.

The *Sakamoto et al.* [2016] parameterizations were particularly sensitive to mass emissions flux and fire area, as well as wind speed and transport time. If mass emissions flux is interpreted as surrogate for plume concentration, this agrees with our conclusion that increased

plume concentration results in a larger size distribution.

**4. Conclusions**

We characterized the physical and optical properties of 19 aged biomass burning events observed at the Mt. Bachelor Observatory in the summer of 2015. Regional (Northern California and Southwestern Oregon) and Siberian events were observed. Our main conclusions were:

- $\Delta\sigma_{scat}/\Delta CO$ ($\sigma_{scat}$ at STP) enhancement ratio ranged from 0.48-1.29 $Mm^{-1}$ $ppbv^{-1}$, with the majority of events being between 0.8 and 1.25 $Mm^{-1}$ $ppbv^{-1}$.

- Siberian-influenced events had significantly higher $\Delta\sigma_{abs}/\Delta CO$ and MAE, and lower $\omega$ compared to regional events. We propose this is due to MBO sampling the portion on Siberian smoke that has been lofted to higher elevation through pyro-convection, thereby
preferentially sampling emissions of strong flaming combustion conditions. In general flaming conditions produce more BC, which would explain the amplified absorption in the Siberian events.

- Absorption Ångström exponent values were significantly lower for the Siberian events than regional events, which indicates lack of BrC produced by the Siberian fires or loss of
BrC during transport through photobleaching, volatilization, and aerosol-phase reactions.

- Mass scattering efficiencies ranged from 2.50-4.76 $m^2$ $g^{-1}$. MSE was positively correlated with $D_{pm}$ ($R^2 = 0.73$), which agrees with Mie Theory.

- Aerosol number size distribution $D_{pm}$ and $\sigma_g$ ranged from 138-229 nm and 1.53-1.89, respectively. Five of the nineteen events had bimodal distributions, the rest being
unimodal. The unimodal distributions had a prominent "tail" of small-diameter particles (30-90 nm). No distinction could be made between regional and Siberian size distributions.

## 4. Author Contribution

James R. Laing performed the data analysis and prepared the manuscript with assistance
from all co-authors.

## 5. Acknowledgments

Funding for research at MBO was supported by the National Science Foundation (grant number: 1447832). MBO is also supported by a grant from the NOAA Earth System Research Laboratory. The views, opinions and findings contained in this report are those of the author(s)

and should not be construed as an official National Oceanic and Atmospheric Administration or

U.S. Government position, policy or decision. The authors gratefully acknowledge the NOAA

Air Resources Laboratory (ARL) for the provision of the HYSPLIT transport model used in this

publication. The CALIPSO satellite products were supplied from the NASA Langley Research

Center.

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

**Table 1.** Identified BB plumes at MBO during the summer of 2015. All enhancement ratios are obtained by taking the slope of a RMA linear regression between the two species. ND ("no data") indicates missing data. WC in the MAE column signifies a weak correlation ($R^2 < 0.60$).

| Event number | Event date and time (UTC) | Event duration (hours) | Source fire location | $\Delta$WV (g/kg) | $\Delta\sigma_{scat}/\Delta$CO (Mm$^{-1}$ ppbv$^{-1}$) | $\Delta\sigma_{abs}/\Delta$CO (Mm$^{-1}$ ppbv$^{-1}$) | MSE (m$^2$ g$^{-1}$) | MAE (m$^2$ g$^{-1}$) | AAE (467-660 nm) | $\omega$ (528 nm) | $D_{pm}$ (nm) | $\sigma_g$ |
|---|---|---|---|---|---|---|---|---|---|---|---|---|
| 1 | 7/31/15 15:35-17:10 | 1.58 | OR | 0.16 | 1.13 | 0.036 | ND | ND | 3.15 | 0.97 | 164 | 1.72 |
| 2 | 8/9/15 2:55-8:55 | 6 | CA, OR | 1.62 | 0.89 | WC | 3.17 | 0.085 | 3.45 | 0.98 | 138 | 1.82 |
| 3 | 8/9/15 13:35-8/10/15 0:00 | 10.42 | CA, OR | 2.07 | 1.24 | 0.033 | 3.29 | 0.087 | 3.72 | 0.98 | 156 | 1.7 |
| 4 | 8/10/15 1:10-5:55 | 4.75 | CA, OR | 1.86 | 1.05 | 0.03 | 3.78 | 0.108 | 3.86 | 0.97 | 182 | 1.54 |
| 5 | 8/10/15 6:05-11:40 | 5.58 | CA, OR | 1.25 | 1.09 | 0.034 | 3.44 | 0.106 | 4.02 | 0.97 | 183 | 1.61 |
| 6 | 8/10/15 11:45-14:35 | 2.83 | CA, OR | 1.32 | 0.94 | WC | 3.27 | WC | 4.12 | 0.99 | 177 | 1.61 |
| 7 | 8/10/15 14:40- 8/11/15 6:15 | 15.58 | CA, OR | 1.83 | 1.17 | 0.032 | 3.64 | 0.098 | 3.52 | 0.98 | 186 | 1.62 |
| 8 | 8/11/15 14:20-18:45 | 4.42 | CA, OR | 1.11 | 1.07 | 0.029 | 2.5 | 0.066 | 2.74 | 0.98 | 160 | 1.78 |
| 9 | 8/14/15 10:00-15:35 | 5.58 | OR | 1.12 | 0.48 | 0.007 | 2.75 | 0.042 | 3.06 | 0.99 | 165 | 1.67 |
| 10 | 8/17/15 0:05-3:55 | 3.83 | Siberia | -0.87 | 1.39 | 0.078 | ND | ND | 2.48 | 0.95 | 176 | 1.57 |
| 11 | 8/17/15 17:15- 8/18/15 7:00 | 13.75 | Siberia | -0.22 | 1.06 | 0.060 | ND | ND | 2.5 | 0.95 | 179 | 1.69 |
| 12 | 8/18/15 16:05 - 8/19/15 16:40 | 24.58 | Siberia | 0.56 | 1.29 | 0.075 | ND | ND | 2.3 | 0.95 | 196 | 1.64 |
| 13 | 8/19/15 17:40 - 8/20/15 3:05 | 9.42 | Siberia | 0.6 | 1.12 | 0.052 | ND | ND | 2.25 | 0.96 | 175 | 1.76 |
| 14 | 8/22/15 15:30-18:05 | 2.58 | Siberia | -3.1 | 1.97 | 0.078 | 4.76 | 0.188 | 3.59 | 0.96 | 229 | 1.73 |
| 15 | 8/23/15 3:55-7:00 | 3.08 | Siberia | -2.45 | 1.09 | 0.059 | 2.84 | 0.156 | 2.51 | 0.96 | 162 | 1.89 |
| 16 | 8/23/15 9:50 - 8/25/15 6:50 | 45 | CA, OR | 1 | 1.13 | 0.029 | 4.06 | 0.107 | 3.15 | 0.98 | 205 | 1.58 |
| 17 | 8/25/15 12:45 - 8/26/15 19:00 | 30.25 | CA, OR | 1.67 | 0.88 | 0.027 | 3.75 | 0.111 | 3.12 | 0.98 | 181 | 1.6 |
| 18 | 8/26/15 7:15 - 8/28/15 11:15 | 40 | CA, OR | 1.35 | 0.89 | 0.031 | 3.7 | 0.128 | 3.48 | 0.97 | 191 | 1.53 |
| 19 | 8/28/15 17:40 - 8/29/15 6:15 | 12.58 | CA, OR | 1.54 | 0.69 | ND | 2.94 | ND | ND | ND | 164 | 1.58 |
| Regional BB events (mean ± stdev) | | | | 1.38 ± 0.49 | 0.97 ± 0.21 | 0.03 ± 0.01 | 3.36 ± 1.03 | 0.09 ± 0.04 | 3.45 ± 1.04 | 5.71 ± 1.65 | 170 ± 15.7 | 1.67 ± 0.08 |
| Siberian BB events (mean ± stdev) | | | | -0.91 ± 1.56 | 1.32 ± 0.34 | 0.07 ± 0.01 | 3.8 ± 2.05 | 0.17 ± 0.09 | 2.61 ± 0.49 | 4.16 ± 0.6 | 181 ± 19.7 | 1.77 ± 0.1 |

$\Delta$WV is water vapor enhancement, calculated for each event by subtracting the average WV for the summer sampling period from the WV value at the time when maximum CO was observed.

Aerosol scattering $\sigma_{scat}$ (550 nm) and absorption $\sigma_{abs}$ (528 nm) measurements were converted to STP.

MSE and MAE calculated as the $\Delta\sigma_{scat}/\Delta$PM1 and $\Delta\sigma_{abs}/\Delta$PM1 enhancement ratios, respectively.

$D_{pm}$ is the geometric mean diameter and $\sigma_g$ is the geometric standard deviation of the SMPS aerosol size distribution.

WC indicates a weak correlation in the MAE column ($R^2 < 0.60$).

ND indicates missing data. PM data was not available for events 1 and 10-13; absorption data was not available for events 19 and 20.

**Figures:**

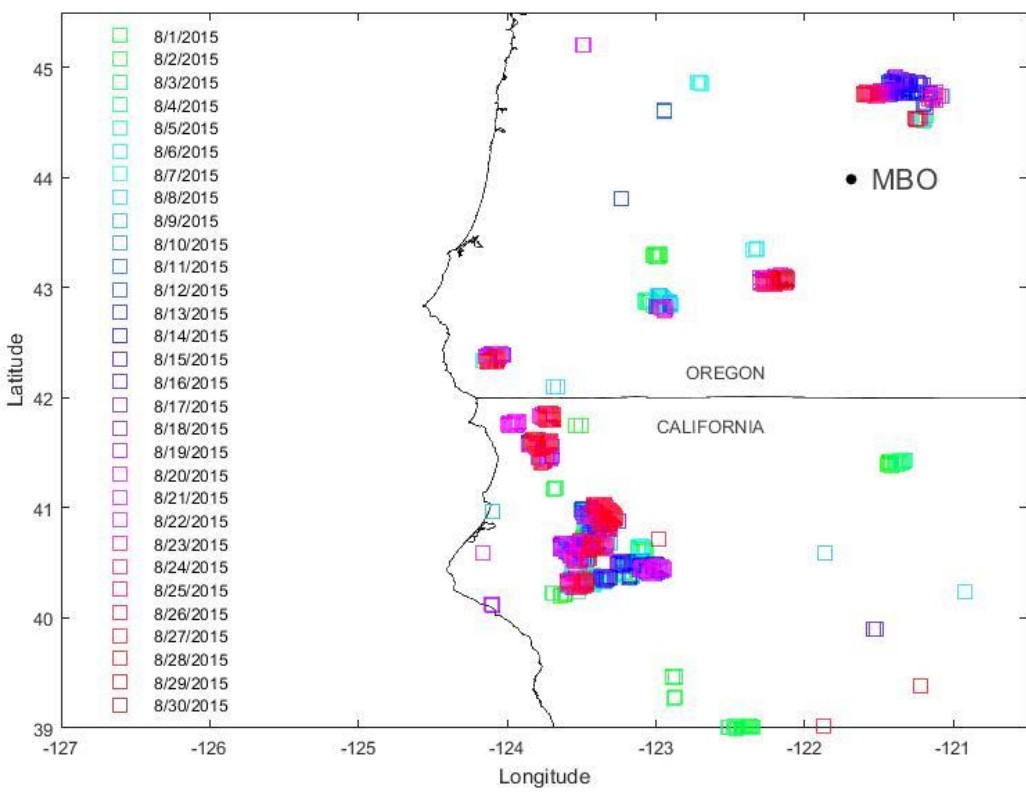

**Figure 1.** MBO and MODIS firespots colored by date for the month of August.


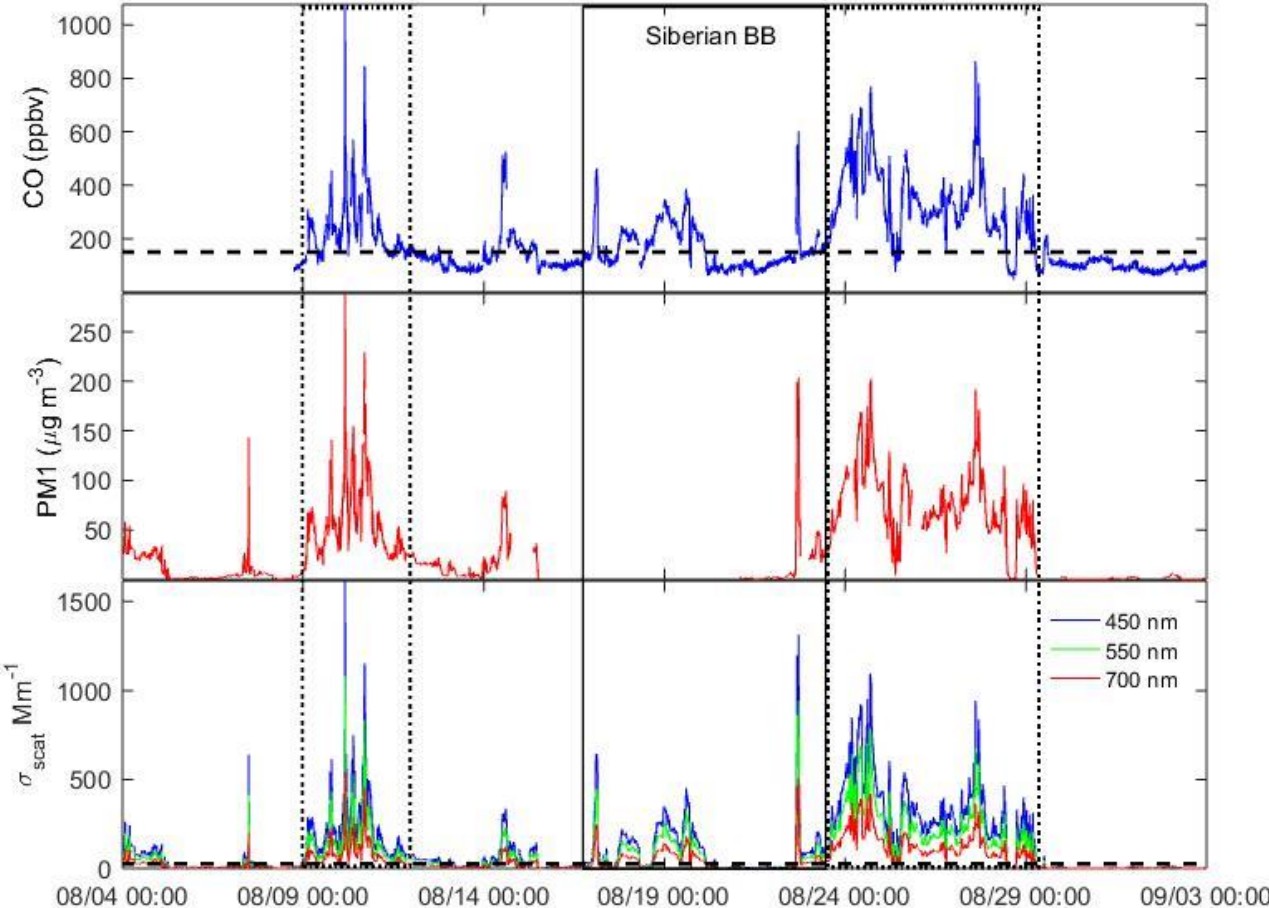

**Figure 2.** Time series of CO, PM1, and aerosol scattering ($\sigma_{scat}$) at MBO during August. Threshold values (dashed black lines) used for BB event criteria are displayed for CO (150 ppbv) 685 and scattering (20 Mm$^{-1}$). The dotted boxes represent multi-day periods of regional BB and encompasses events 2-8 and 16-19, respectively. The solid box represents the period influenced by Siberian BB and encompasses events 10-15.

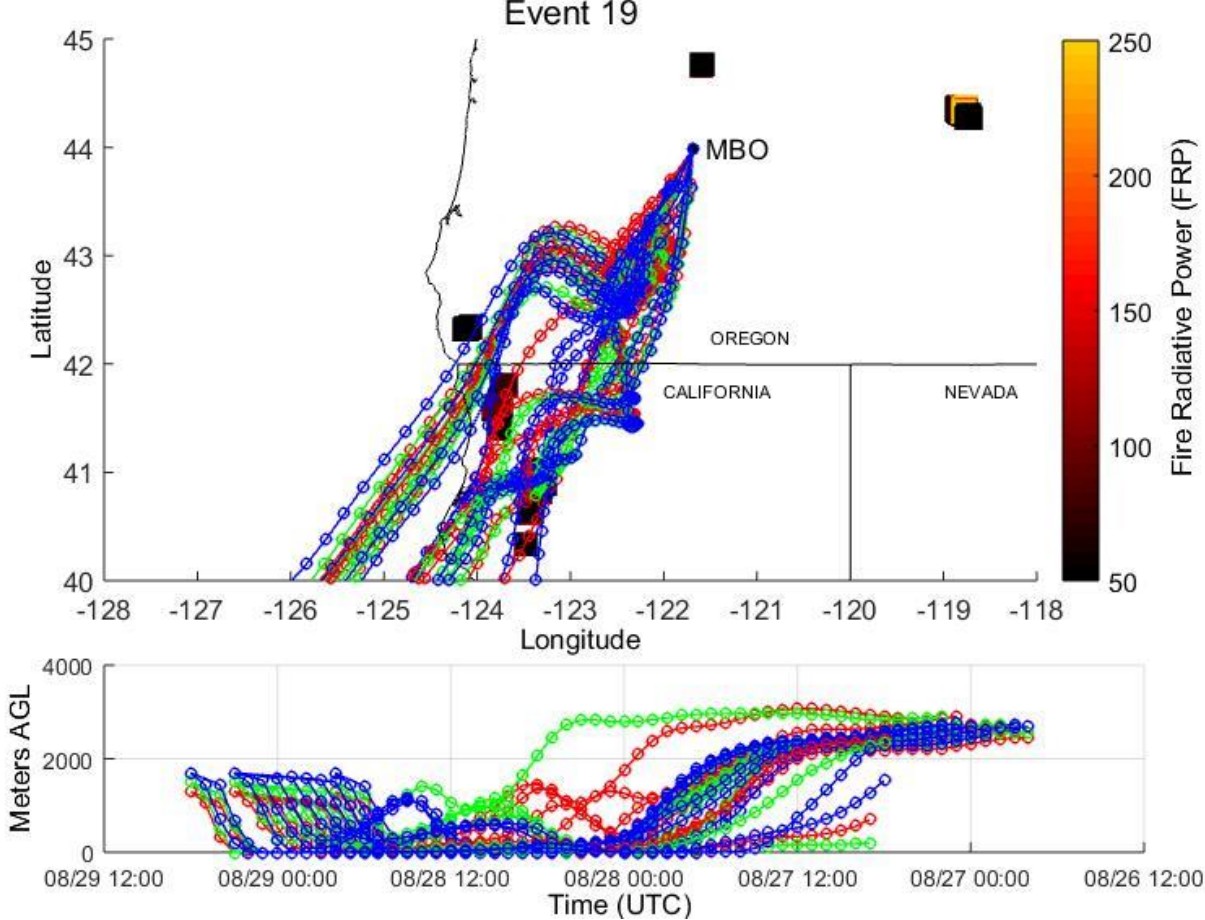

**Figure 3.** Hysplit Back-trajectories for Event 19. The blue back-trajectories have a starting height of 1700 m amgl (above model ground level), the green a starting height of 1500 m amgl, and the red a starting height of 1300 m amgl. The squares are MODIS fire spots from 8/27/2015-8/29/2015 and are colored based on their fire radiative power (FRP).

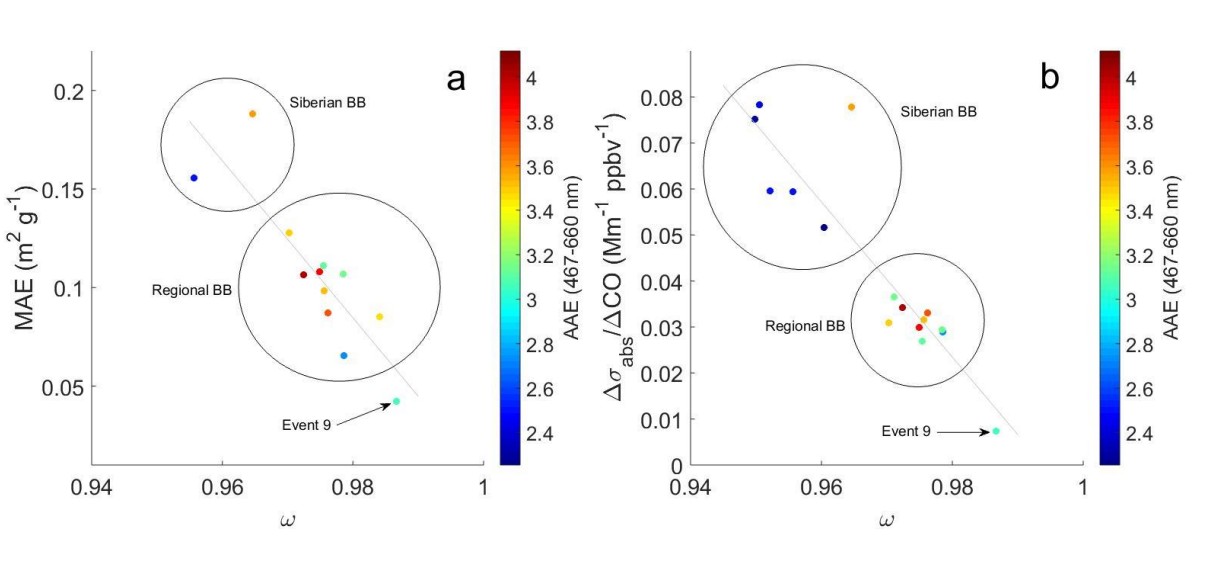


**Figure 4.** Scatter plots of (a) mass absorption efficiency (MAE) and (b) absorption enhancement ratio $\Delta\sigma_{abs}/\Delta CO$ versus single scattering albedo ($\omega$). MAE values were not calculated for four of the six Siberian-influenced events due to missing PM1 data.


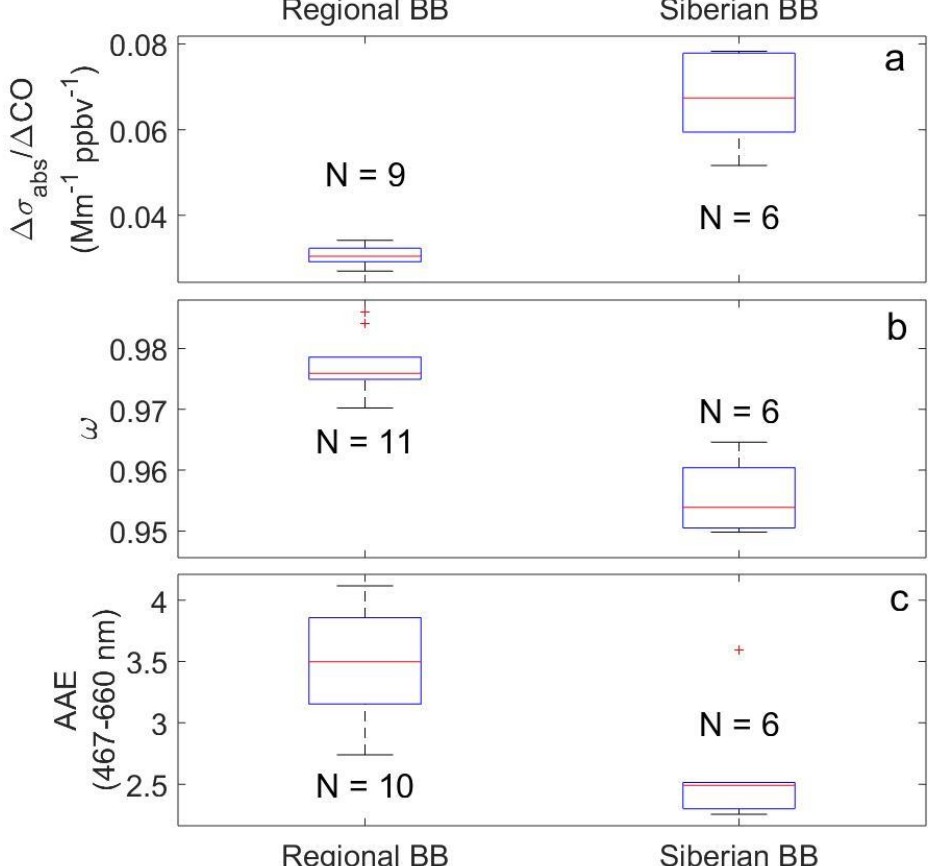

**Figure 5.** Boxplots of (a) $\Delta\sigma_{abs}/\Delta CO$, (b) single scattering albedo ($\omega$) measured at 528 nm, and (c) Absorption Ångström exponent (AAE) for absorption measurements at 467 and 660 nm for regional BB events and Siberian influenced events. N indicates the number of events for each
box. Lower and upper whiskers represent the minimum and maximum values, respectively. Lower and upper lines of the box represent the 25th and 75th percentiles, respectively. The red line in the middle of the box represents the median, and the red plus mark represents outliers.


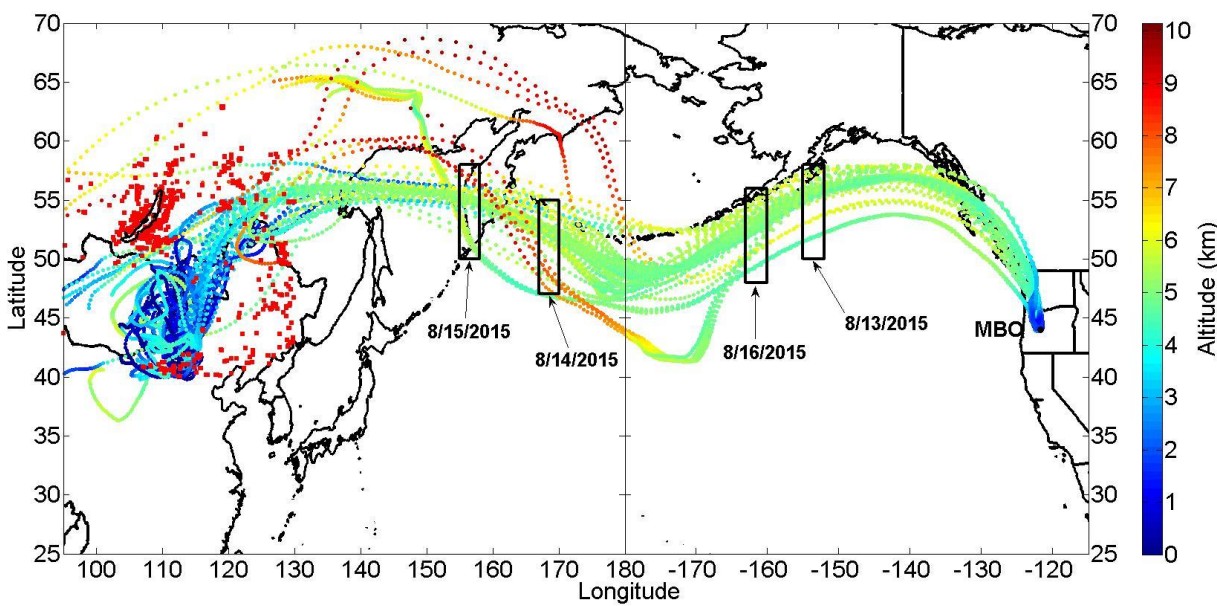

**Figure 6**. Most of the HYSPLIT back-trajectories for Siberian events (events 10-15) plotted as a function of altitude. Roughly 10% of the back-trajectories that did not follow the main transport track were not plotted. Forest fires from 8/7/2015 to 8/16/2015 identified by the Fire INventory from NCAR (FINN) fires are marked by red squares. These transects are not sequential and do not track one plume of Siberian smoke, but rather illustrate the extensive eastward transport of Siberian smoke over the course of the week. The four black boxes represent the locations of smoke plumes identified by CALIPSO cross sections detailed in Figures S1-S4.

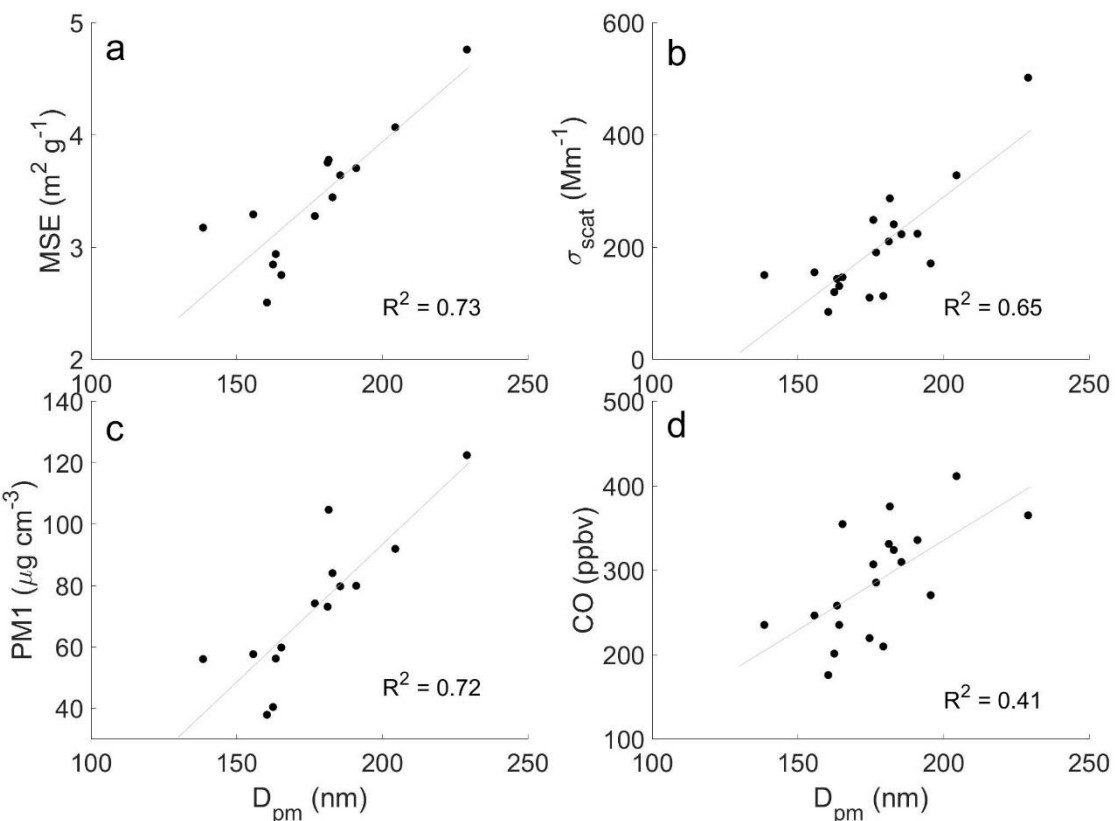

**Figure 7.** Scatter plots of (a) MSE, (b) $\sigma_{scat}$, (c) PM1, and (d) CO versus $D_{pm}$ for the BB events at MBO in the summer of 2015.


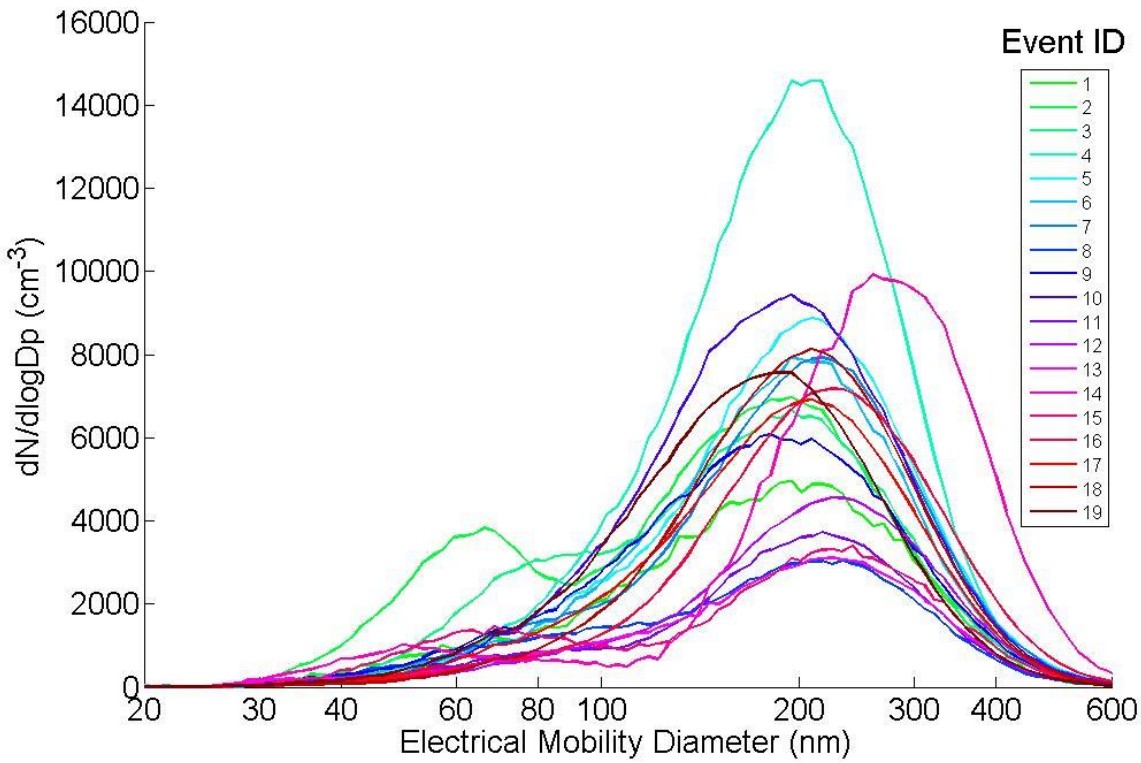

**Figure 8.** Event integrated aerosol number size distributions (corrected to STP) in dN/dlogDp (# cm$^{-1}$).