# Peer review of "Physical and Optical Properties of Aged Biomass Burning Aerosol from Wildfires in Siberia and the Western US at the Mt. Bachelor Observatory"

_Atmospheric Chemistry and Physics, 2016_

## Referee Comment (RC1) · Anonymous Referee #1 · 7 Aug 2016

Summary

The paper summarizes the physical and optical properties of aerosols from biomass smoke from regional to continental scale events. The paper is appropriate, well-focused and should eventually be publishable in ACP. I recommend the following minor modifications and additional analysis prior to publication.

Technical Comments

Techniques and analysis seem sound. The criteria for smoke impacts and differentiating LRT and regional smoke events with water vapor seems well-thought out.

Certainly the trend is consistent of lower SSA for the Siberian fires and thus a flaming,

higher MCE fire. However, all the SSA values are all relatively high suggesting an MCE on the lower end of the range (mixed to smoldering combustion). This is worth commenting. For reference see the Liu paper below.

The analysis brought to mind a recent paper by the CSU group examining emitted and aged biomass smoke sizing and radiative properties paper referenced below which may provide a useful intercomparison and context.

Table 1 is useful, however would be more useful with the following additions:

o A summary mean +/- s.d. for the regional versus Siberian events, maybe 2 lines at the bottom

o Adding in your rough estimate of the age of the plume for each case which was stated as a range elsewhere. Do the size distributions with Aitken modes correspond to the younger plumes? Are there any other conclusions to be drawn?

Figure 4. I'm not sure how the percentiles are done with such small numbers of samples, symbol with whiskers showing the range seems more appropriate. You're really comparing the Siberian to regional fires, why separate into 3 groups? I could only see one small outlier symbol on the chart.

Figure 5. Meaning of this? The events symbols are not distinguishable; I would simply delineate Siberian vs. regional with different symbols and colors. With the exception of CO, these parameters are by definition or calculation interdependent. Is the take home message something along the lines of, "Biomass smoke events as indicated by elevated CO concentrations featured shifts to larger sizes driving higher PM mass concentration, light scattering coefficients, and the highest overall mass scattering efficiencies."

Mechanics and Presentation

The presentation is appropriate in terms of length, style and diction. Figures are appropriate.

[Figure]

Why put the hysplit trajectories plot in supplementary material though? The CALISPO images are appropriately in the supplement. However, the paper is short enough it can accommodate the additional figure rather than the annoyance of looking elsewhere.

I noted a few inconsistencies (line 158 and 196 for example) in variable, citation, and subscript italics, check throughout.

Line 109 "was located prior to any...." Aerosol instrumentation?

Line 133, I recommend breaking out as an equation rather than inline.

Line 192, I believe you mean Period 2.

Line 209, "ascended from the boundary layer (BL) to...." MBO?

Line 242 superscript missing

Line 280 "hygroscopy" replace with hygroscopicity

Line 299 "Mei" replace with Mie

Line 375 "preformed" replace with performed

Liu, S., et al. (2014), Aerosol single scattering albedo dependence on biomass combustion efficiency: Laboratory and field studies, Geophys. Res. Lett., 41, 742–748, doi:10.1002/2013GL058392.

Rapidly evolving ultrafine and fine mode biomass smoke physical properties: Comparing laboratory and field results, JOURNAL OF GEOPHYSICAL RESEARCH: ATMOSPHERES, Volume 121, Issue 10, 27 May 2016, Pages: 5750–5768

---

## Referee Comment (RC2) · Anonymous Referee #2 · 2 Sep 2016

This manuscript characterizes the physical and optical properties of biomass burning aerosols transported over the Mt. Bachelor Observatory during the summer 2015. This is an important dataset and deserves to be published. This being said, I feel the analysis of the measurement data could have been better processed with appropriate uncertainty values assigned. Hence, I would recommend publication of this manuscript after mandatory revision. Below are my major comments: 1) The fact that the authors observe a low single scattering albedo and Absorption Angstrom exponent implies majority of the aerosols were black carbon (BC) and not Brown Carbon. This is corroborated by higher MCE values indicating flaming phase of combustion. So, my question is: why are the authors surprised at lack of BrC aerosols? BrC aerosols are generated

from smoldering fire phase, mostly associated with peat burning. Smoldering phase is associated with very low MCE, which was not observed in this study. What the authors observed were over crown forest fires (flaming phase). This concept has to be made clear in the text and the abstract. Otherwise, the confusion that only BC is generated from Siberian forest fires would propagate in the community. 2) The abstract and the text says "aerosol light scattering and absorption" were measured. Please specify what parameters were measured—scattering and absorption cross-sections or coefficients? I am assuming the authors measure coefficients. 3) The scattering and absorption coefficients were adjusted to desired wavelengths using Ansgtrom exponents calculated by other studies. Could the authors specify the values used to extrapolate? 4) Reading Fisher et al (2010), it seems the SAE values ranged between 2-2.8? What's the rationale behind using this range? Why not use 4 instead? Since all particles are in Rayleigh regime (sub-micron), their scattering cross-sections decrease in power-law exponents of 4 with increasing wavelength. So, why did the authors adopt SAE of $\sim$2.4 and not 4? 5) Figure 5 doesn't make any sense to me. Could the authors provide any physical explanation behind the correlations? Scattering in the Rayleigh regime goes as square of particle volume, which probably explains the poor correlation. But what about the others. If one cannot explain or even hypothesize the reason behind a plot, why put it. I suggest the authors to remove this unnecessary plot from the main manuscript or move it to Supplementary Materials. 6) Please provide an error analysis of the techniques used to measure absorption and scattering coefficients. Uncertainties involved during calculation of SAE, AAE using previously published data should be mentioned. A paragraph on error analysis is a must for this kind of study. I would further suggest to propagate these values to the error bars in figure 4. 7) The manuscript has grammatical and typographical errors. I suggest a thorough editing done to the contents during its revision.

---

## Author Comment (AC1) · 14 Oct 2016

Response to reviewer #1:
We thank the reviewer for their comments on the article. We've responded to the individual comments below.

_Summary_
_The paper summarizes the physical and optical properties of aerosols from biomass smoke from regional to continental scale events. The paper is appropriate, well focused and should eventually be publishable in ACP. I recommend the following minor modifications and additional analysis prior to publication._

_Technical Comments_

_Techniques and analysis seem sound. The criteria for smoke impacts and differentiating LRT and regional smoke events with water vapor seems well-thought out._

_Certainly the trend is consistent of lower SSA for the Siberian fires and thus a flaming, higher MCE fire. However, all the SSA values are all relatively high suggesting an MCE on the lower end of the range (mixed to smoldering combustion). This is worth commenting. For reference see the Liu paper below._
Response: This is a good point as all of the SSA's we observed were >0.95, but low SSA values seem to be only observed in studies of primary emissions. Vakkari et al. (2014) and Yokelson et al. (2009) observed SSA to increase significantly with aging in wildland fire plumes within an hour of emission. In addition, at MBO Briggs et al. (2016) found that BB plumes do not follow the SSA vs MCE parameterization from Liu et al. (2014) and that all SSA values were >0.92 despite very high MCE values. These high SSA values were attributed to SOA formation and increased scattering efficiency driven by particle growth. Given this and that fact that MCE couldn't be accurately derived due to the high CO2 background, I'm hesitant to say that the equate the relatively high SSA values for the Siberian plumes, which have been atmospherically processed for up to 10 days, to mixed to smoldering combustion.
I have added a paragraph to address this in Section 3.3 (Lines 302-315).

_The analysis brought to mind a recent paper by the CSU group examining emitted and aged biomass smoke sizing and radiative properties paper referenced below which may provide a useful intercomparison and context._
Response: The Carrico et al. (2016) paper is a useful paper to compare with our paper. We cited it in comparison with our results for size distributions in Line 388.

_Table 1 is useful, however would be more useful with the following additions:_
_- A summary mean +/- s.d. for the regional versus Siberian events, maybe 2 lines at the bottom_
Response: This is a good idea and was added to Table 1.

_- Adding in your rough estimate of the age of the plume for each case which was stated as a range elsewhere. **Do the size distributions with Aitken modes correspond to the younger plumes? Are there any other conclusions to be drawn?**_

Response: We looked into this but it was too ambiguous to draw any definitive conclusions. Most of the regional BB events were influenced by multiple fires which made it nearly impossible to properly estimate plume age. Additional language to explain this was added in Section 3.2 (Lines 256-260), and Figure 3 was added for a visual explanation.

As far as the events with bimodal distributions, 3 of the 5 are of Siberian origins. Given this we suspect the Aitken mode in these size distributions most likely represents a secondary source from within the boundary layer. We have put this in Line 371.

*Figure 4. I'm not sure how the percentiles are done with such small numbers of samples, symbol with whiskers showing the range seems more appropriate. You're really comparing the Siberian to regional fires, why separate into 3 groups? I could only see one small outlier symbol on the chart.*

Response: Figure 4 (now Figure 5) was changed to 2 groups, Siberian and Regional fires. Due to the outlier for the Siberian BB AAEs we feel the box plots give a more accurate portrayal of the distribution on the values rather than a mean and whiskers showing the range

*Figure 5. Meaning of this? The events symbols are not distinguishable; I would simply delineate Siberian vs. regional with different symbols and colors. With the exception of CO, these parameters are by definition or calculation interdependent. Is the take home message something along the lines of, "Biomass smoke events as indicated by elevated CO concentrations featured shifts to larger sizes driving higher PM mass concentration, light scattering coefficients, and the highest overall mass scattering efficiencies."*

Response: Figure 5 was removed and replaced with Figure 7, which is a similar plot but with $D_{pm}$ on the x-axis. This looks at the dependence of different variables on size distribution instead of MSE. For Figure 7 all the events were changed to the same color and not individually identified since $D_{pm}$ was shown not to depend on transport time.

The previous plots showed a correlation between MSE and $\sigma_{scat}$, PM1, and CO, all of which can be thought of as surrogates for plume concentration. Since $\sigma_{scat}$, PM1, and CO are all correlated with size distribution (Figure 7), the correlations with MSE are likely just a function of particle size and cause no causal effect. In summary it seems the more concentrated the BB plume (higher $\sigma_{scat}$, PM1, CO) the larger the size distribution, which in turn increases the MSE. Given this, we decided to highlight the correlation between $\sigma_{scat}$, PM, CO versus Dpm instead of MSE.

Language regarding the new figure is in Lines 350-357, and 390-398.

*Mechanics and Presentation*

*The presentation is appropriate in terms of length, style and diction. Figures are appropriate.*

*Why put the hysplit trajectories plot in supplementary material though? The CALISPO images are appropriately in the supplement. However, the paper is short enough it can accommodate the additional figure rather than the annoyance of looking elsewhere.*

Response: This Figure was moved to the main manuscript. It is now Figure 6.

*I noted a few inconsistencies (line 158 and 196 for example) in variable, citation, and subscript italics, check throughout.*

Response: The manuscript was checked for inconsistencies and proofread.

Line 109 "was located prior to any. . .." Aerosol instrumentation?
Response: This was changed

 Line 133, I recommend breaking out as an equation rather than inline.
Response: This was changed

Line 192, I believe you mean Period 2.
Response: This was changed

Line 209, "ascended from the boundary layer (BL) to. . .." MBO?
Response: This was changed

Line 242 superscript missing
Response: This was changed

Line 280 "hygroscopy" replace with hygroscopicity Line 299 "Mei" replace with Mie
Response: This was changed

Line 375 "preformed" replace with performed
Response: This was changed

Liu, S., et al. (2014), Aerosol single scattering albedo dependence on biomass combustion efficiency: Laboratory and field studies, Geophys. Res. Lett., 41, 742–748, doi:10.1002/2013GL058392.

Carrico (2016) - Rapidly evolving ultrafine and fine mode biomass smoke physical properties: Comparing laboratory and field results, JOURNAL OF GEOPHYSICAL RESEARCH: ATMOSPHERES, Volume 121, Issue 10, 27 May 2016, Pages: 5750–5768

---

## Author Comment (AC2) · 14 Oct 2016

Response to reviewer #2:
We thank the reviewer for their comments on the article. We've responded to the individual comments below.

This manuscript characterizes the physical and optical properties of biomass burning aerosols transported over the Mt. Bachelor Observatory during the summer 2015. This is an important dataset and deserves to be published. This being said, I feel the analysis of the measurement data could have been better processed with appropriate uncertainty values assigned. Hence, I would recommend publication of this manuscript after mandatory revision. Below are my major comments:

1) The fact that the authors observe a low single scattering albedo and Absorption Angstrom exponent implies majority of the aerosols were black carbon (BC) and not Brown Carbon. This is corroborated by higher MCE values indicating flaming phase of combustion. So, my question is: why are the authors surprised at lack of BrC aerosols? BrC aerosols are generated from smoldering fire phase, mostly associated with peat burning. Smoldering phase is associated with very low MCE, which was not observed in this study. What the authors observed were over crown forest fires (flaming phase). This concept has to be made clear in the text and the abstract. Otherwise, the confusion that only BC is generated from Siberian forest fires would propagate in the community.
Response:
We explain why we suspect there is a lack of BrC in the Siberian events in Section 3.3 (Lines 316-330). We explain that the lower AAE values could be due to a lack of BrC initially produced by the fires due to flaming conditions. We also suggest that the low AAEs could be due to the loss of BrC during transport through photobleaching, volatilization, and aerosol-phase reactions. We were not able to calculate MCE due to the long transport times, low CO2 enhancements and high background. We do state in the abstract (Line 23), results (Lines 279-283), and conclusions (Lines 422-427) that we suspect the Siberian events represent a selective portion of the fire plume that is more likely to represent flaming conditions.

2) The abstract and the text says "aerosol light scattering and absorption" were measured. Please specify what parameters were measured, scattering and absorption cross-sections or coefficients? I am assuming the authors measure coefficients.
Response: We measured scattering and absorption coefficients. We have clarified this in Line 21, 89, 106, 118, and 203.

3) The scattering and absorption coefficients were adjusted to desired wavelengths using Ansgtrom exponents calculated by other studies. Could the authors specify the values used to extrapolate? Reading Fisher et al (2010), it seems the SAE values ranged between 2-2.8? What's the rationale behind using this range? Why not use 4 instead? Since all particles are in Rayleigh regime (sub-micron), their scattering cross-sections decrease in power-law exponents of 4 with increasing wavelength. So, why did the authors adopt SAE of 2.4 and not 4?
Response: The SAE values were calculated for each 5-min average using the scattering coefficient measurements at 450 nm and 550 nm. We then used this SAE for each 5-min period to adjust the scattering coefficient measurement at 550 nm to 528 as per equation 1. This was

done so we could calculate single scattering albedo from scattering and absorption measurements at the same wavelength. This has been clarified in the manuscript in Lines 142-152.

4) Figure 5 doesn't make any sense to me. Could the authors provide any physical explanation behind the correlations? Scattering in the Rayleigh regime goes as square of particle volume, which probably explains the poor correlation. But what about the others. If one cannot explain or even hypothesize the reason behind a plot, why put it. I suggest the authors to remove this unnecessary plot from the main manuscript or move it to Supplementary Materials.
Response: Figure 5 was removed and replaced with Figure 7, which is a similar plot but with $D_{pm}$ on the x-axis. This looks at the dependence of different variables on size distribution instead of MSE. For Figure 7 all the events were changed to the same color and not individually identified since $D_{pm}$ was shown not to depend on transport time.
The previous plots showed a correlation between MSE and $\sigma_{scat}$, PM1, and CO, all of which can be thought of as surrogates for plume concentration. Since $\sigma_{scat}$, PM1, and CO are all correlated with size distribution (Figure 7), the correlations with MSE are likely just a function of particle size and cause no causal effect. In summary it seems the more concentrated the BB plume (higher $\sigma_{scat}$, PM1, CO) the larger the size distribution, which in turn increases the MSE. Given this, we decided to highlight the correlation between $\sigma_{scat}$, PM, CO versus Dpm instead of MSE.
Language regarding the new figure is in Lines 350-357, and 390-398.

5) Please provide an error analysis of the techniques used to measure absorption and scattering coefficients. Uncertainties involved during calculation of SAE, AAE using previously published data should be mentioned. A paragraph on error analysis is a must for this kind of study. I would further suggest to propagate these values to the error bars in figure 4.
Response: An error analysis was completed. Precision and total uncertainties were calculated for all of the optical measurements and provided in Table S1. We added description of the error analysis in the Methods Section. Lines 212-128 for aerosol scattering, Lines 135-142 for aerosol absorption, Line 156 for AAE, Lines 185-198 for enhancement ratios.

6) The manuscript has grammatical and typographical errors. I suggest a thorough editing done to the contents during its revision.
Response: We went through the paper and corrected any grammatical and typographical errors. We also checked for consistency.